# Snow-induced buffering in aerosol–cloud interactions

Takuro Michibata[1], Kentaroh Suzuki[2], and Toshihiko Takemura[1]

[1]Research Institute for Applied Mechanics, Kyushu University, Fukuoka 816-8580, Japan
[2]Atmosphere and Ocean Research Institute, The University of Tokyo, Chiba 277-8568, Japan

**Correspondence:** Takuro Michibata (michibata@riam.kyushu-u.ac.jp)

**Abstract.** Complex aerosol–cloud–precipitation interactions lead to large differences in estimates of aerosol impacts on climate among general circulation models (GCMs) and satellite retrievals. Typically, precipitating hydrometeors are treated diagnostically in most GCMs, and their radiative effects are ignored. Here, we quantify how the treatment of precipitation influences the simulated effective radiative forcing due to aerosol–cloud interactions ($\mathrm{ERF_{aci}}$) using a state-of-the-art GCM with a two-moment prognostic precipitation scheme that incorporates the radiative effect of precipitating particles, and investigate how microphysical process representations are related to macroscopic climate effects. Prognostic precipitation substantially weakens the magnitude of $\mathrm{ERF_{aci}}$ (by approximately 54%) compared with the traditional diagnostic scheme, and this is the result of the increased longwave (warming) and weakened shortwave (cooling) components of $\mathrm{ERF_{aci}}$. The former is attributed to additional adjustment processes induced by falling snow, and the latter stems largely from riming of snow by collection of cloud droplets. The significant reduction in $\mathrm{ERF_{aci}}$ does not occur without prognostic snow, which contributes mainly by buffering the cloud response to aerosol perturbations through depleting cloud water via collection. Prognostic precipitation also alters the regional pattern of $\mathrm{ERF_{aci}}$, particularly over northern mid-latitudes where snow is abundant. The treatment of precipitation is thus a highly influential controlling factor of $\mathrm{ERF_{aci}}$, contributing more than other uncertain "tunable" processes related to aerosol–cloud–precipitation interactions. This change in $\mathrm{ERF_{aci}}$ caused by the treatment of precipitation is large enough to explain the existing difference in $\mathrm{ERF_{aci}}$ between GCMs and observations.

## 1 Introduction

Aerosols play significant roles in the climate system (Twomey, 1977; Albrecht, 1989) by modifying the radiation budget (aerosol–radiation interactions; ARI) and the hydrological cycle through interactions with clouds (aerosol–cloud interactions; ACI). Quantitative estimates of anthropogenic aerosol forcing, however, are still largely uncertain (Boucher et al., 2013) because of the complex interactions among aerosols, clouds, and climate across wide spatiotemporal scales (Mülmenstädt and Feingold, 2018). Reducing these uncertainties associated with the effect of aerosol forcing on climate is one of the most challenging issues in climate science (Seinfeld et al., 2016).

A key uncertainty arises from the complex response of clouds to aerosol perturbations (Wang et al., 2012). Clouds are considered to respond to perturbed aerosols in two opposing ways; i.e., the so-called "cloud lifetime" effect (Albrecht, 1989) and the "buffered system" effect (Stevens and Feingold, 2009), in a regime-dependent manner (Wood, 2012; Michibata et al., 2016). The cloud water susceptibility to aerosols depends strongly upon cloud type (Christensen et al., 2016), as well as

ambient environmental conditions (Toll et al., 2019), which results in non-monotonic cloud responses (Gryspeerdt et al., 2019) and therefore diverse impacts on climate (Chen et al., 2014).

These observational findings are also supported by process modeling studies using large-eddy simulations (Lebo and Feingold, 2014; Seifert et al., 2015). General circulation models (GCMs), however, show a large spread in cloud susceptibility to aerosols (Ghan et al., 2016; Zhang et al., 2016), and tend to overestimate the magnitude of ACI compared with satellite retrievals (Malavelle et al., 2017). This means that current GCMs are not able to reproduce the buffering of cloud responses to aerosol perturbations (Jing et al., 2019). Aerosol-induced radiative forcing at the top of the atmosphere (TOA) that includes rapid adjustments caused by ACI, termed effective radiative forcing ($\mathrm{ERF_{aci}}$), varies widely among GCMs (Shindell et al., 2013; Zelinka et al., 2014). This results in a "best estimate" of global annual mean $\mathrm{ERF_{aci}}$ of $-0.45\,\mathrm{W\,m^{-2}}$ with a 90% confidence interval of $-1.2$ to $0.0\,\mathrm{W\,m^{-2}}$ (Boucher et al., 2013), as reported in the fifth assessment report of the Intergovernmental Panel on Climate Change (IPCC AR5). This uncertainty range has remained large (e.g., Bellouin et al., 2020) since the early IPCC reports.

As a consequence of the challenges described above, GCMs tend to show more negative $\mathrm{ERF_{aci}}$ than that inferred from satellite retrievals (Quaas et al., 2009; Chen et al., 2014) even though retrieval errors (Ma et al., 2018) are considered (Michibata and Suzuki, 2020). This suggests that current GCMs may be missing a compensating warming effect caused by aerosols. The "missing warming" in GCMs may be solved by taking aerosol effects on (i) deep convective clouds (Wang et al., 2011) and (ii) mixed-phase clouds (Lohmann and Hoose, 2009) into consideration, as these effects can modify the ice microphysics due to aerosols and also lead to an adjustment in the longwave component (Lohmann, 2017). A recent multi-model analysis (Heyn et al., 2017) demonstrated that simpler GCMs that parameterize the aerosol effect on liquid-phase clouds alone have negligibly small longwave ERF, whereas more sophisticated GCMs that include microphysical adjustments of ice- and mixed-phase clouds as well as liquid-phase clouds produce larger magnitude ERF values for both the terrestrial ($\mathrm{ERF^{LW}}$) and solar ($\mathrm{ERF^{SW}}$) components. The changes to $\mathrm{ERF^{LW}}$ and $\mathrm{ERF^{SW}}$ were found to nearly cancel each other out and result in a net ERF ($\mathrm{ERF^{Net}}$) of a magnitude that is similar to that generated by the simpler GCMs. The robustness of this near cancelation, however, largely depends on how microphysical processes in ice- and mixed-phase clouds, which are typically much more complex than in liquid-phase clouds (Lohmann, 2017), are represented in GCMs.

Among these processes, precipitation processes involving falling hydrometeors (i.e., rain and snow) are particularly simplified in current GCMs, which is likely to lead to nonnegligible uncertainty in $\mathrm{ERF_{aci}}$ (Gettelman, 2015). In general, precipitation is treated diagnostically in GCMs (hereinafter "DIAG"), with precipitation being immediately removed from the atmosphere within a single model timestep. This overweights autoconversion relative to accretion to produce precipitation (Posselt and Lohmann, 2008), which results in the pronounced sensitivity of cloud water to aerosols because autoconversion is the only process that directly depends on aerosols (Gettelman et al., 2013). Snow also has significant effects on collection processes among other hydrometeors (Sant et al., 2015), as well as on atmospheric circulation (Li et al., 2014). However, snow-induced impacts on $\mathrm{ERF_{aci}}$ are much less understood (Waliser et al., 2011) because extremely limited GCMs incorporate prognostic precipitation with the radiative effects of falling hydrometeors (see discussion in Michibata et al. (2019)).

This study investigates this unexplored area of ACI, with a particular focus on precipitation (rain and snow) processes and their impacts on $\text{ERF}_{\text{aci}}$, and with the goal of advancing our understanding of the fundamental linkage of microphysical process representations to their macroscopic climate effects. For this purpose, we use a recently developed global aerosol–climate model, MIROC6-SPRINTARS (Tatebe et al., 2019), which is implemented with a two-moment prognostic precipitation scheme (hereinafter "PROG") that includes the radiative effects of precipitation (Michibata et al., 2019). Through a comparison with the traditional DIAG scheme, we use the PROG-scheme model to identify the source of discrepancies in $\text{ERF}_{\text{aci}}$ between GCMs and satellite observations that are related to precipitation processes. A suite of sensitivity experiments is also performed with the model to isolate the relative contributions of different microphysical processes to $\text{ERF}_{\text{aci}}$ and to quantify how uncertainties inherent in these processes translate to $\text{ERF}_{\text{aci}}$ uncertainty. This single-model approach has the advantage of not being affected by varying physics representations, as in the case of multi-model analysis (cf. Materials and Methods).

## 2 Materials and methods

### 2.1 MIROC6-SPRINTARS aerosol–climate model

We used version 6 of the global aerosol–climate model, MIROC6-SPRINTARS (Tatebe et al., 2019) in this work. The aerosol module, SPRINTARS (Takemura et al., 2009), predicts the mass mixing ratios of the main aerosol species in the troposphere (black carbon, organic matter, sulfate, soil dust, and sea salt) and gas-phase precursors of sulfate (sulfur dioxide and dimethyl sulfide) and organic matter (terpene and isoprene). The cloud microphysics are based on the prognostic probability density function (PDF) scheme, which represents the subgrid-scale variability of temperature and total water content (Watanabe et al., 2009), and is coupled to an ice microphysics scheme (Wilson and Ballard, 1999). The model treats cloud water and ice using a two-moment representation, by prognosing both mass and number mixing ratios (Takemura et al., 2009). Cloud droplet nucleation is represented by a Köhler-theory-based parameterization (Abdul-Razzak and Ghan, 2000). Note that although the standard version of MIROC6-SPRINTARS uses Berry's autoconversion parameterization (Berry, 1968), results presented in this paper apply an alternative formulation based on Khairoutdinov and Kogan (2000), which is used in the PROG version (Michibata et al., 2019) for a robust comparison (described later). The default MIROC6-SPRINTARS model treats precipitation diagnostically, and its radiative effect is not considered.

We also used another version of the model that employs a prognostic precipitation framework (Michibata et al., 2019). This version prognoses mass and number mixing ratios for both rain and snow, as well as cloud liquid and ice condensates (full two-moment scheme). Microphysical processes are calculated iteratively by using sub-time steps (60 s), except for the sedimentation of precipitation, which can be shorter subject to the vertical CFL criteria. The PROG scheme considers the radiative effect of precipitating hydrometeors. The particle shapes of solid hydrometeors are prescribed by assuming hexagonal columns for cloud ice and dendrite crystals for snow bulk categories, which correspond to elements of a radiation table (Yang et al., 2013). For more details, please refer to the model description for the latest version of MIROC6 (Tatebe et al., 2019; Michibata et al., 2019).

## 2.2 Experimental setup

We performed sets of simulations with different aerosol emissions for the years 2000 (present-day; PD) and 1850 (pre-industrial; PI). All simulations used prescribed climatological sea surface temperature and sea ice. Simulations were integrated for 6 years, with the last 5 years being used in the subsequent analysis. The model resolution was T85L40 (ca. 1.4° resolution in longitude and latitude with 40 vertical levels), and the standard model timestep was 12 min. The modeled cloud cover and its horizontal distribution (Fig. S1) are in good agreement with CALIPSO-GOCCP satellite data (Chepfer et al., 2010) in PROG but underestimated in DIAG, which were evaluated using the COSP2 satellite simulator package (Swales et al., 2018) using an additional full one-year run under the PD conditions.

Additional sensitivity experiments were performed, by replacing the precipitation framework, changing the liquid autoconversion scheme, and masking ice microphysics and aerosol freezing processes (discussed later in Sect. 4). To quantify how the treatment of precipitation influences the simulated $\mathrm{ERF_{aci}}$, two experiments; i.e., one that incorporates a prognostic treatment of rain but not snow (PRDS), and another that applies the full prognostic version (PROG), were compared with the default simulation with diagnostic precipitation (DIAG). To evaluate the snow radiative effect, a pair of simulations with and without snow radiation were also carried out using the PROG framework. For liquid microphysics, four commonly used autoconversion schemes (BE68 (Berry, 1968); BE94 (Beheng, 1994); LD04 (Liu and Daum, 2004); and SB06 (Seifert and Beheng, 2006)) were compared with the default PROG simulation using the KK00 scheme (Khairoutdinov and Kogan, 2000). Results from the sensitivity experiments which were adjusted by a factor of 0.1 for the Wegener–Bergeron–Findeisen (WBF) process (Wegener, 1911; Bergeron, 1935; Findeisen, 1938), aggregation, riming efficiency, and freezing ratios of homo- and heterogeneous nucleation, were subtracted from the default PROG result to quantify the impact of the targeted process. In this study, $\mathrm{ERF_{aci}}$ is defined as the change in net cloud radiative forcing at the TOA under clean-sky (Ghan, 2013) with fixed ocean conditions, but allows atmospheric processes including rapid adjustments in the response to aerosol changes, from PI to PD (Boucher et al., 2013).

If needed, these experiments were retuned so that the imbalance of the radiative flux at the TOA remained within $1.0\,\mathrm{W\,m^{-2}}$. Model tuning was conducted by modifying scale factor for accretion rate but not autoconversion for warm rain process because the latter can influence the magnitude of ACI due to the direct relation to droplet number (Michibata and Takemura, 2015; Jing et al., 2019) and thus the precipitation initiation (Mülmenstädt et al., 2020). This is effective for modifying SW radiation, but if needed, cloud ice and snow processes were also tuned for modifying LW radiation by changing scale factors for the fall speed of hydrometeors, which may be uninfluential on $\mathrm{ERF_{aci}}$ because they are not involved directly in the hydrometeor number densities.

## 3  Weakening of $\mathrm{ERF_{aci}}$ with prognostic precipitation

Figure 1 compares geographical distributions of $\mathrm{ERF_{aci}}$ simulated by the DIAG and PROG models. In DIAG, a strong negative $\mathrm{ERF_{aci}}$ is observed over East Asia, Europe, and North America where anthropogenic pollution dominates. This is attributed to the cloud lifetime effect caused by anthropogenic aerosols, which increases low warm clouds and hence shortwave reflectance.

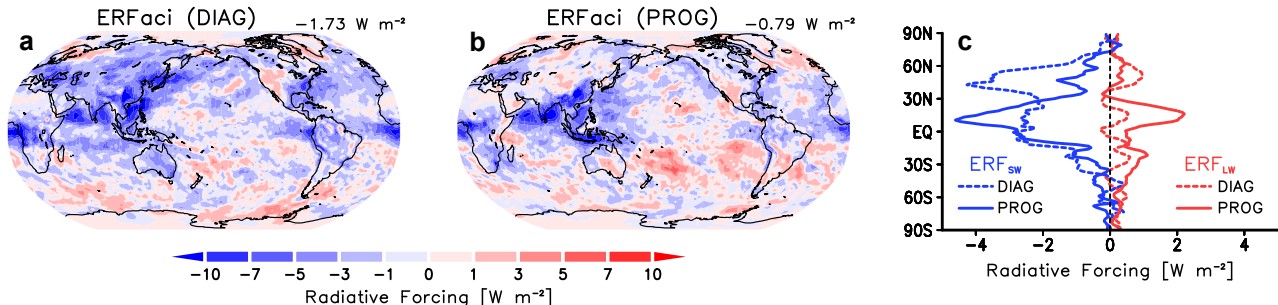

**Figure 1.** Geographical distribution of the annual mean clean-sky $\mathrm{ERF_{aci}}$ for the **(a)** DIAG and **(b)** PROG precipitation schemes. $\mathrm{ERF_{aci}}$ is decomposed into (red) longwave and (blue) shortwave components in the **(c)** zonal mean field for the (dashed) DIAG and (solid) PROG schemes.

The global annual mean $\mathrm{ERF_{aci}}$ reaches $-1.7\,\mathrm{W\,m^{-2}}$, which is outside the bound of the uncertainty range ($-1.2\,\mathrm{W\,m^{-2}}$) in IPCC AR5. The geographical pattern is consistent with other GCMs (Shindell et al., 2013; Zelinka et al., 2014).

In PROG, however, the majority of the strong negative forcing over anthropogenic regions is reduced significantly, resulting in a reduction of around 54% in global-mean $\mathrm{ERF_{aci}}$. Although the geographical pattern is somewhat different from previous reports using other GCMs (discussed in the next section), the global mean $\mathrm{ERF_{aci}}$ ($-0.8\,\mathrm{W\,m^{-2}}$) is much closer to satellite-based estimates (Chen et al., 2014; Christensen et al., 2016, 2017; Douglas and L'Ecuyer, 2020). The total aerosol ERF associated with ARI and ACI ($\mathrm{ERF_{ari+aci}}$) in PROG ($-1.1\,\mathrm{W\,m^{-2}}$) is only half that generated by DIAG ($-2.1\,\mathrm{W\,m^{-2}}$).

This significant reduction in $\mathrm{ERF_{aci}}$ in PROG results from a substantial weakening of $\mathrm{ERF_{aci}^{SW}}$ particularly over mid-latitudes of the Northern Hemisphere, and enhanced warming of $\mathrm{ERF_{aci}^{LW}}$ over low latitudes in both hemispheres (Fig. 1c). The zonal distribution shows that stronger (weaker) $\mathrm{ERF_{aci}^{LW}}$ accompanies stronger (weaker) $\mathrm{ERF_{aci}^{SW}}$, which is in line with Heyn et al. (2017). To understand the impact of precipitation treatment on $\mathrm{ERF_{aci}}$, decompositions of global mean $\mathrm{ERF_{aci}}$ into its SW and LW components are shown for alternate configurations of precipitation in MIROC6 (Fig. 2). Figure 2 confirms that the significant reduction of $\mathrm{ERF_{aci}}$ in PROG is contributed to by both increased $\mathrm{ERF_{aci}^{LW}}$ and weakened $\mathrm{ERF_{aci}^{SW}}$, in stark contrast to previous CMIP5 model results (Heyn et al., 2017) in which cloud-ice-induced changes to $\mathrm{ERF^{SW}}$ and $\mathrm{ERF^{LW}}$ cancel each other out to result in few net ERF changes within the DIAG framework. This difference in the present study from previous results is attributed to the snow-induced modulation of ACI newly incorporated into our model.

The impact of snow on ACI can be understood in more detail using the results shown in Fig. 2, which includes two intermediate versions of PROG; i.e., one that incorporates prognostic rain but diagnostic snow (PRDS) to isolate the relative impacts of rain vs snow on $\mathrm{ERF_{aci}}$, and one that represents prognostic rain and snow but without the radiative effects of snow (OFF/SnwRad). Regarding the LW component, the global mean $\mathrm{ERF_{aci}^{LW}}$ of PROG ($+0.7\,\mathrm{W\,m^{-2}}$) is more than twice as large as those of DIAG ($+0.2\,\mathrm{W\,m^{-2}}$) and PRDS ($+0.3\,\mathrm{W\,m^{-2}}$). The OFF/SnwRad simulation also shows weaker $\mathrm{ERF_{aci}^{LW}}$ relative to the standard PROG simulation (Fig. 2). These results suggest that the warming LW effect comes mainly from adjustments induced by snow together with its radiative effects, in addition to cloud-ice effects included in CMIP5 models as well as our

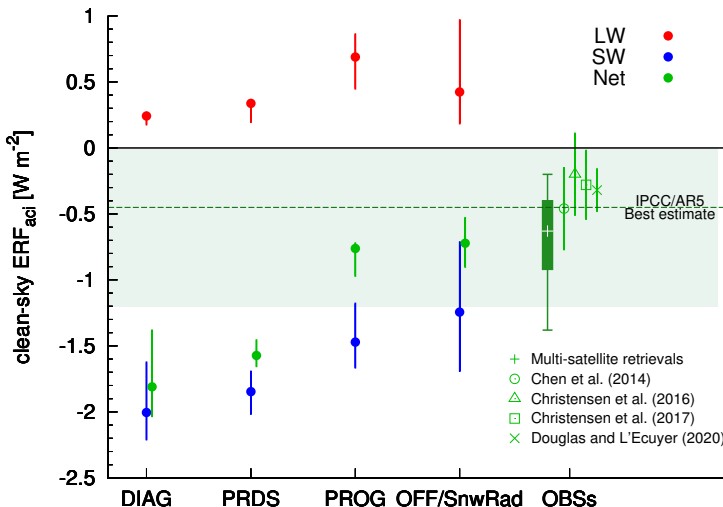

**Figure 2.** $ERF_{aci}$ ($ERF_{aci}^{Net}$ in green; $ERF_{aci}^{LW}$ in red; $ERF_{aci}^{SW}$ in blue) simulated from MIROC6 with different precipitation frameworks. The $ERF_{aci}^{Net}$ values from observation-based studies (Chen et al., 2014; Christensen et al., 2016, 2017; Douglas and L'Ecuyer, 2020) and their probable range (box-whisker) calculated by correcting the effect of retrieval limitations (Michibata and Suzuki, 2020) based on Ma et al. (2018) are also shown. Error bars and plots in MIROC6 represent the minimum/maximum and median of the interannual variability, respectively. Shaded in light-green is the uncertainty range of $ERF_{aci}$ estimated from IPCC AR5 (Boucher et al., 2013). The prognostic rain with diagnostic snow scheme is denoted as "PRDS". The sensitivity experiment without snow radiative effects is denoted as "OFF/SnwRad".

model. The $ERF_{aci}^{LW}$ is significant over the Indian Ocean and Southeast Asia (not shown), which is also similar to the other
model, CAM5-MARC-ARG (Grandey et al., 2018). This is attributable to the increased ice nuclei (IN) due to biomass burning for example, partly supporting the convective invigoration (Rosenfeld et al., 2014) although GCMs do not have enough capability to resolve the convective cloud systems. The increased IN results in a faster glaciation and thus enhances snowfall due to the WBF process (i.e., glaciation indirect effect). These mixed- and ice-phase microphysical processes are more elaborated in the PROG scheme, and the associated LW change induced by snow is incorporated only in the PROG, which contributes to
the higher $ERF_{aci}^{LW}$ across the globe.

The PROG scheme also reduces the SW component ($ERF_{aci}^{SW}$) relative to DIAG, particularly over anthropogenic regions (Fig. 1b) in the Northern Hemisphere mid-latitudes. A well-known mechanism for the reduction in $ERF_{aci}^{SW}$ is the enhancement of accretion with a smaller contribution from autoconversion, as in PROG (not shown), with only the latter process depending upon the cloud droplet number concentration ($N_c$) (Posselt and Lohmann, 2008). The smaller contribution of autoconversion
in PROG mitigates the excessive cloud water susceptibility to aerosols that occurs in DIAG models (Gettelman et al., 2015; Michibata et al., 2019). However, Fig. 2 shows that the replacement in liquid-phase precipitation alone from DIAG to PRDS cannot explain the significant reduction of $ERF_{aci}^{SW}$ from DIAG to PROG, suggesting that ice-phase processes involving falling snow influence the magnitude of $ERF_{aci}^{SW}$, as discussed in the next section.

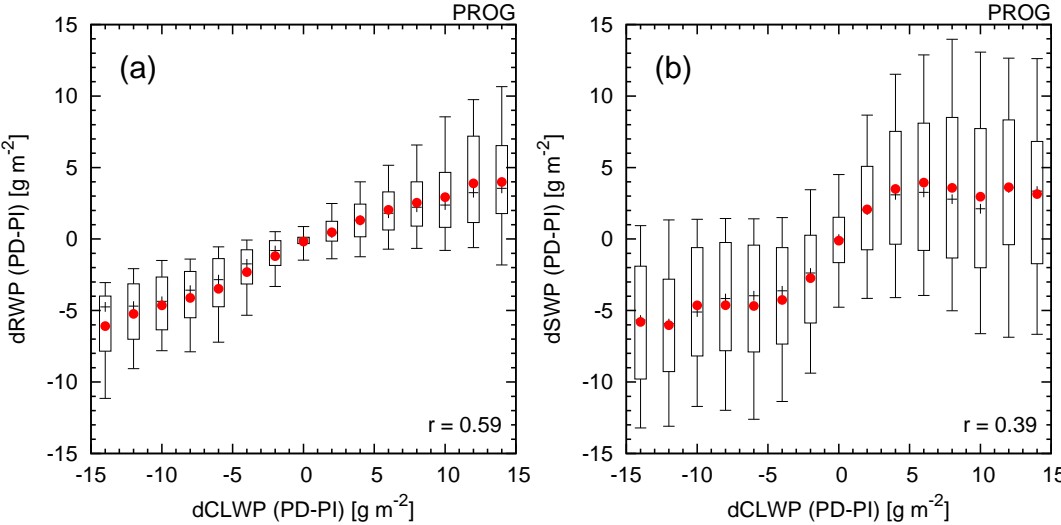

**Figure 3.** Relationship between the change in annual mean CLWP and that in annual mean **(a)** RWP and **(b)** SWP, from the change in aerosols from PI to PD conditions, simulated using the PROG scheme. Box-whisker plots represent the 10th, 25th, 50th (black "+"), 75th, and 90th percentiles of the data within each bin based on the annual mean. Plots in red show the mean. The correlation coefficient ($r$) is given in the figure.

This reduction of $\mathrm{ERF_{aci}^{SW}}$ in PROG relative to DIAG is also impossible to explain by the response of cloud ice alone,
because cloud ice should increase $\mathrm{ERF_{aci}^{SW}}$ towards more negative values because of aerosol-induced increases in cloud optical thickness. This is indeed what is happening with the DIAG framework in the CMIP5 multi-model results (Heyn et al., 2017), in which models with aerosol effects on cloud ice (not snow) show much stronger $\mathrm{ERF_{aci}^{SW}}$, which is large enough to cancel the enhancement of $\mathrm{ERF_{aci}^{LW}}$. In contrast, our PROG model reduces $\mathrm{ERF_{aci}^{SW}}$. We hypothesize that the prognostic treatment of snow plays an important role in weakening $\mathrm{ERF_{aci}^{SW}}$ through microphysical processes involving cloud water and snow, as
discussed below.

## 4  Relationship of microphysics and $\mathrm{ERF_{aci}}$

Next, we discuss the role of prognostic precipitation in determining $\mathrm{ERF_{aci}}$ by addressing the following two questions raised in the previous section:

1.  *Why does the geographical pattern of* $\mathrm{ERF_{aci}}$ *in PROG differ from that of DIAG?*

2.  *Why does the prognostic treatment of snow effectively weaken* $\mathrm{ERF_{aci}^{SW}}$ *?*

To this end, we first explore how precipitating hydrometeors can modulate the cloud water susceptibility to perturbed aerosols. Figure 3 shows how the change in the cloud liquid water path (CLWP) relates to changes in precipitating hydrometeor paths; i.e., the rainwater path (RWP) and the snow water path (SWP), through pre-industrial (PI) to present-day (PD) changes

in aerosols. The PD minus PI change (susceptibility) in RWP is highly correlated ($r = 0.59$) with that in CLWP (Fig. 3a). We interpret this strong correlation to be the result of the close co-variance of cloud and rainwater through aerosol perturbations, with the cloud water being a direct source of the rainwater. The PD minus PI change in SWP is also positively correlated, though weaker ($r = 0.39$), than that in CLWP (Fig. 3b), suggesting that precipitating snow also co-varies with cloud water through aerosol perturbations. Given that SWP is significantly larger than RWP in our model (Fig. S2; see also Michibata et al. (2019)), and that snowflakes, with residence times longer than those of rain, are more likely to interact with clouds, the increased CLWP caused by anthropogenic aerosols can act as an efficient source of snow via interactions among cloud droplets and snowflakes (e.g., riming), likely resulting in the evident robust positive relationship.

These positive correlations between precipitating hydrometeors and cloud water suggest that aerosol-induced increases in cloud mass are caused, in part, by increases of rain and snow in PROG, in contrast to those caused by increases of cloud water and ice alone in DIAG. Given that raindrops and snowflakes are optically much thinner in the SW spectrum than cloud droplets and ice crystals, respectively, increases of precipitating hydrometeors can explain both the stronger $\mathrm{ERF}_{\mathrm{aci}}^{\mathrm{LW}}$ and weaker $\mathrm{ERF}_{\mathrm{aci}}^{\mathrm{SW}}$ in PROG compared with DIAG (Figs. 1 and 2). Furthermore, falling snow is more likely to deplete underlying cloud droplets in PROG, with its explicit representation of the riming process, which can lead to a reduction of cloud water susceptibility to aerosols. This proposed mechanism can also explain the systematic change in the geographical distribution of $\mathrm{ERF}_{\mathrm{aci}}$ between DIAG and PROG (Fig. 1). Indeed, regions with a significant reduction in $\mathrm{ERF}_{\mathrm{aci}}$ (i.e., over East Asia, Europe, and North America) correspond well to those with large values of SWP (Fig. S3), where the PD – PI increase in CLWP is also reduced significantly (Fig. S4). These results lend further credence to the hypothesis of snow-induced buffering of ACI in our model.

The buffering process, via interactions among hydrometeors described above, depends strongly on the fundamental uncertainty in model representations of various microphysical processes. We therefore now further explore how $\mathrm{ERF}_{\mathrm{aci}}$ and its buffering by precipitation processes are sensitive to microphysical process representations as summarized in Fig. 4 (see also Sect. 2.2 for details of experiments). The processes examined here are the autoconversion of liquid droplets, the Wegener–Bergeron–Findeisen (WBF) process, the aggregation of ice crystals, riming, and ice nucleation by freezing aerosols, which are all important sources of uncertainty in GCMs (Lawson and Gettelman, 2014; Gettelman, 2015; Sant et al., 2015). As expected, the simulated $\mathrm{ERF}_{\mathrm{aci}}$ is highly sensitive to the autoconversion scheme used, mainly because of its varying dependence on $N_c$ among the various schemes (Jing et al., 2019). Different liquid autoconversion scheme with PROG can change $\mathrm{ERF}_{\mathrm{aci}}$ by 39%, from $-18\%$ to $+21\%$ (blue bars in Fig. 4). The impacts of the autoconversion scheme on $\mathrm{ERF}_{\mathrm{aci}}$, however, are smaller than those of the treatment of rain and snow (ca. 54% change).

The mixed- and ice-phase processes (WBF, aggregation, and riming), represented more explicitly with a larger degree of freedom in PROG than in DIAG, can change $\mathrm{ERF}_{\mathrm{aci}}$ by 15%, from $-13\%$ to $+2\%$ (cyan bars in Fig. 4). Among the mixed- and ice-phase microphysics processes, the process found to most influence $\mathrm{ERF}_{\mathrm{aci}}$ is the riming of cloud droplets on snow, supporting the hypothesized mechanism of snow-induced buffering of ACI discussed above. The magnitude of $\mathrm{ERF}_{\mathrm{aci}}$ is sensitive to ice nucleation processes as well (green bars in Fig. 4), because the change in ice number concentration directly controls the size of the crystals and thus the conversion timescale from ice to snow in our model. Although the $\mathrm{ERF}_{\mathrm{aci}}$

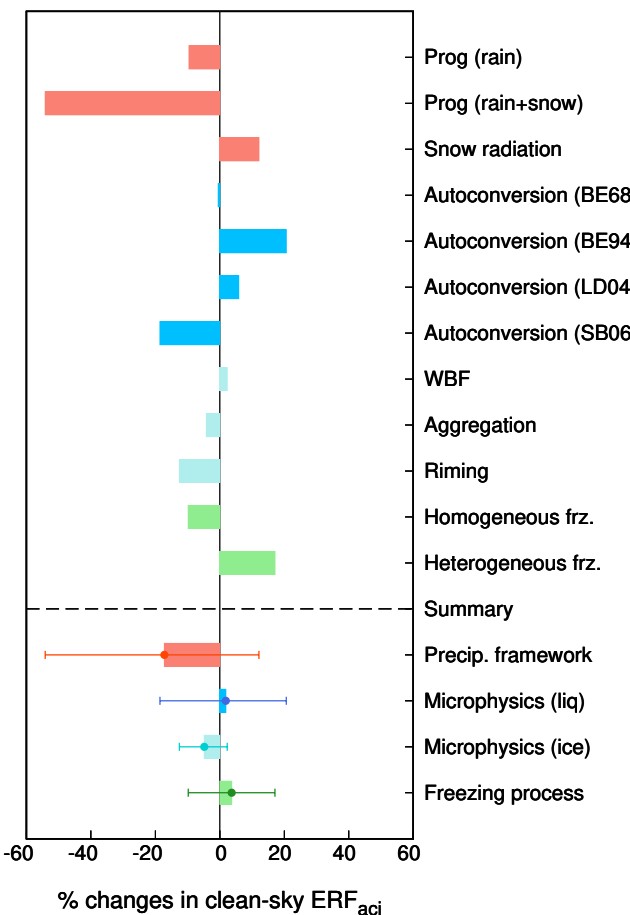

**Figure 4.** Percentage change of global annual mean clean-sky $ERF_{aci}$ in response to (red) the precipitation treatment, (blue) liquid microphysics, (cyan) ice microphysics, and (green) nucleation of new ice particles due to freezing. Error bars represent the minimum and maximum range for each component considered in this study.

variations with changing liquid and ice microphysical processes do not reach the difference of $ERF_{aci}$ between the DIAG and
PROG (i.e., 54%), both wet-scavenging of aerosols and coalescence scavenging of cloud droplets also contribute to the ACI reduction (McCoy et al., 2020) due to the accretion-driven buffering mechanisms (Michibata and Suzuki, 2020), which should explain the remaining part of the $ERF_{aci}$ difference.

In summary, we found that the treatment of precipitation (PROG vs DIAG) is the most influential factor controlling $ERF_{aci}$ (red bars in Fig. 4) among all of the "tunable knobs" associated with the various microphysical processes in our model. It
should also be emphasized that the $ERF_{aci}$ change caused by the precipitation treatment (ca. 54% in magnitude), absent from previous climate modeling studies, has the potential to resolve some of the differences between satellite estimates of $ERF_{aci}$ (Bellouin et al., 2013; Chen et al., 2014; Christensen et al., 2017) and GCMs (Shindell et al., 2013; Zelinka et al., 2014; Heyn

et al., 2017). These findings need to be tested further using other GCMs as they incorporate prognostic precipitation in future studies.

## 5   Summary and future work

In this study, the sensitivities of $\mathrm{ERF_{aci}}$ to various treatments of precipitation and microphysical process representations in a GCM have been systematically examined. As few GCMs incorporate explicit representations of two-moment prognostic precipitation with the radiative effects of precipitating hydrometeors (e.g., CAM6 MG2/MG3 (Gettelman et al., 2015, 2019), E3SM (Rasch et al., 2019), GISS-E3, and MIROC6 CHIMERRA (Michibata et al., 2019)), we used a single model framework to evaluate the sensitivities. This also allowed us to avoid uncertainties from inter-model differences in parameterizations other than the targeted processes.

We found that the treatment of precipitation in GCMs (PROG vs DIAG) has a significant impact on the magnitude of $\mathrm{ERF_{aci}}$ (Figs. 1 and 2), which we interpret to be driven mainly by collection processes among precipitating snow and cloud droplets (i.e., riming). As the SWP is more than twice as large as the RWP in our PROG model, and is in good agreement with satellite retrievals (Michibata et al., 2019), falling snowflakes efficiently accrete and deplete the underlying cloud water thus partly cancelling the CLWP response to aerosols. Changes in RWP and SWP though PI to PD aerosol perturbations were also positively correlated with that in CLWP (Fig. 3), suggesting that snow can co-exist with cloud water to a degree sufficient to buffer the cloud water response to aerosol perturbations (Fig. 4). The signatures of the snow-induced buffering are also found geographically over regions with significant reductions in $\mathrm{ERF_{aci}}$ (e.g., East Asia, Europe, and North America) that correspond closely to regions with particularly large SWP (Figs. 1 and S3). Sets of sensitivity experiments, performed both with and without snow radiative effects, did not reveal a significant difference in $\mathrm{ERF_{aci}}$ as a result of the near cancellation of SW and LW changes caused by snow. This means that the prognostic treatment of precipitation itself is critical for the buffering of ACI. Accordingly, the impact of a prognostic treatment of precipitation on the magnitude of $\mathrm{ERF_{aci}}$ was greater than changes to any of the other "tunable knobs" inherent to the various microphysical processes (e.g., autoconversion, ice microphysics, and ice nucleation). Notably, precipitation-driven buffering effects (ca. 54% change in $\mathrm{ERF_{aci}}$) can broadly explain the current model-observation discrepancy in estimated $\mathrm{ERF_{aci}}$ (Boucher et al., 2013; Lohmann, 2017).

However, the results presented here are based on a single GCM framework and need to be replicated using other GCMs as they incorporate prognostic precipitation frameworks in the future (Li et al., 2020). This is particularly true because little is known about aerosol influences on mixed- and ice-phase clouds as well as deep convective clouds (Rosenfeld et al., 2014; Fan et al., 2018) and cirrus clouds (Penner et al., 2018) at a fundamental process-level, and the degree of microphysical complexity differs widely among GCMs (Heyn et al., 2017). Although the responses of clouds and precipitation to aerosol perturbations are therefore likely to be model dependent, the sign of the response of $\mathrm{ERF_{aci}}$ to the precipitation framework and microphysical processes is consistent with a previous assessment using CAM5/MG2 (Gettelman, 2015), suggesting that the major findings of this study will apply across the models. Thus, it is left for important future studies to quantify the inter-model spread of $\mathrm{ERF_{aci}}$ sensitivity to microphysical processes and their interplay with precipitation processes as more

GCMs begin to include prognostic precipitation. Furthermore, a theoretical approach (Glassmeier and Lohmann, 2016) and idealized process modeling (Glassmeier et al., 2019) are also required urgently to solidify the process-level understanding of snow-induced buffering hypothesis, which are our important future work beyond the present study.

This study primarily focused on $\mathrm{ERF_{aci}}$ sensitivities to the CLWP adjustment rather than cloud fraction adjustment, because aerosol effects are directly linked to the CLWP change through the modification of the mass conversion rate from cloud water to rainwater that itself relates to the treatment of precipitation (i.e., DIAG vs PROG). However, it is important in future studies to separating the ACI into the Twomey forcing and rapid adjustments of CLWP and cloud fraction (e.g., Goren and Rosenfeld, 2014; Mülmenstädt et al., 2019) for better understanding how the treatment of precipitation influences micro- and macroscopic cloud properties (Michibata and Suzuki, 2020), which relates to the fundamental inter-model spread in $\mathrm{ERF_{aci}}$ (Gryspeerdt et al., 2020; Bellouin et al., 2020).

*Data availability.* The results of the MIROC-SPRINTARS simulations used to produce the figures can be obtained from the corresponding author upon reasonable request.

*Author contributions.* TM designed research; TM, KS, and TT performed research; TM analyzed data; and TM and KS wrote the paper.

*Competing interests.* The authors declare that they have no conflict of interest.

*Acknowledgements.* The authors would like to thank the developers of both SPRINTARS and MIROC. The new microphysics and radiation schemes were optimized by Koji Ogochi. Simulations by MIROC-SPRINTARS were executed on the SX-ACE supercomputer system of the National Institute for Environmental Studies, Japan. This study was supported by JSPS KAKENHI Grant Numbers JP18J00301, JP19K14795, and JP19H05669; the Environment Research and Technology Development Fund (JPMEERF20202R03) of the Environmental Restoration and Conservation Agency of Japan; the Integrated Research Program for Advancing Climate Models (TOUGOU) Grant Number JPMXD0717935457 from the Ministry of Education, Culture, Sports, Science and Technology (MEXT); and the Collaborative Research Program of the Research Institute for Applied Mechanics, Kyushu University. The authors are grateful to Johannes Mülmenstädt (Pacific Northwest National Laboratory) and one anonymous reviewer for providing constructive suggestions and comments, that helped to improve the manuscript.

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
