# Peer review of "Snow-induced buffering in aerosol-cloud interactions"

_Atmospheric Chemistry and Physics, 2020_

## Referee Comment (RC1) · Anonymous Referee #1 · 15 May 2020

This manuscript demonstrates the impact of falling hydrometeors on the effective radiative forcing of aerosol-cloud interactions in MIROC6 by modifying their microphysics from diagnostic precipitation to prognostic precipitation. The authors find that the large deviation in ERF(aci) in the old scheme is significantly improved as the magnitude of ERF(aci) is reduced. A series of sensitivity tests reveal that prognostic snow has substantial impact on both shortwave and longwave forcing induced by aerosols. This is an excellent modeling work, particularly that the impacts of many physical processes relating to aerosol-cloud interactions are quantified, as illustrated in Figure 4. However, there are some issues need to be clarified before accepted for publication.

Major concerns:

1. ERF(aci) is determined not only by changes in optical properties of cloud and precip-

itating particles, but also by changes in the cloud cover, particularly in GCM. In the text, the authors only show that the simulated cloud cover is "in good agreement" with the observation (although I think underestimation of high clouds is quite severe). However, the aerosol-induced change in cloud cover is not discussed at all.

2. From this manuscript, it is not clear to me how the aerosol scavenging process works in MIROC6 DIAG or PROG version. I suppose that PROG can simulate aerosol washout more realistic with falling rain and snow, and both ERF(aci) and ERF(ari) could be influenced.

3. This manuscript primarily focuses on the SW, while there is very little discussion about LW. The change in ERF(aci)-LW only attributes to "adjustment induced by snow together with its radiative effect" (Line 139). It is way too brief. The authors should elaborate the physical processes related to the "adjustment".

4. The authors claim that significant reduction in ERF(aci) well corresponds to snow water path (SWP) geographically (Fig. S3). It seems not true for me, particularly over the Southern Ocean. In contrast, the correlation between CLWP (Fig. S4) and ERF(aci) (Fig. 1) looks very high. It may imply that the change in ERF(aci) is primarily determined by CLWP change, and its relation with SWP is not so important.

5. The sequence of inference in Section 4 is complicated and not straightforward to me. The authors fist state that SWP susceptibility is correlated to LWP susceptibility (which I think a little weak) and propose a hypothesis that riming is important to reduce the optical depth of clouds. However, the sensitivity test clearly reveal the importance of riming (Fig. 4). That is, the sensitivity experiment should be considered as the basis of speculation, not the supporting evidence to the hypothesis.

Minor points:

1. It seems to me that Figure 4 only consider the changes in ERF(aci)-SW because snow radiation is to increase ERF here. If it is the case, the figure caption and the

description in the X-axis need to be more clear.

2. Line 87: What is the standard time step in MIROC6 as the sub-time step is 60 s?

3. Line 95: Does it mean PD and PI simulations use the same SST and sea ice?

4. Line 108: WBF should be spelled out.

5. Line 110: ". . . were returned. . . " Does it mean the simulations without TOA balance are not used in analysis?

6. Line 123: This argument regarding ERF(ari) and ERF(aci) is quite indirect. Why not just show the values of these two terms?

7. Line 126: "i.e., close to zero" Its meaning is ambiguous.

8. Line 144: "only the latter process. . ." Should be "former"?

9. Figure 3: the description in X-axis should be CLWP instead of LWP.

10. Line 174: "water mass suspended in the atmosphere" Does it mean cloud only?

11. Line 175: If the total mass is the same, this argument holds. However, in this simulation, the total mass in PROG is larger than DIAG, and then the optical thickness of cloud+precipitation in PROG can be larger than that of cloud only in DIAG.

---

## Referee Comment (RC2) · Johannes Mülmenstädt (Referee) · 29 Jun 2020

I have reviewed "Snow-induced buffering in aerosol–cloud interactions" by Takuro Michibata et al. The authors present an interesting set of sensitivity studies that show that prognostic snowfall in their GCM strongly reduces ERFaci compared to diagnostic snowfall, in large part because the longer residence time of snow leads to greater collection of cloud water by snow, which reduces the relative importance of warm phase ACI. Based on my own work, I think this is a plausible mechanism. There are a few potential weak links in the argument, which I will point out below. I don't think those should hold up publication of this potentially very useful result; after all, no paper is ever the final word on any topic. I recommend minor revisions to clarify the points I list below.

[Figure]

Major points

l. 110: The tuning strategy needs to be described in more detail. The worry with retuning is that the ERFaci difference may not be due to the change that was intentionally made, but due to the retuning.

l. 170 ff.: This is a question rather than a comment. From the conclusion of the paper, I would have expected the relationship between $\Delta$LWP and $\Delta$SWP to be the opposite – that when there is more snow, it would lead to more efficient removal of liquid cloud water. Would you mind making these plots separately for supercooled and non-supercooled water?

Minor points

l. 39 ff.: I don't think this argument is logically consistent. First, the authors say ERFaci in GCMs is "too negative" compared to satellite studies (I would prefer "more negative", since satellite studies have their own problems). But then they cite a (problematic) satellite study with a very negative SW ERFaci to argue that the problem is with the models' LW ERFaci. The rest of the paragraph is fine, but I would suggest removing the first two sentences.

l. 111: Didn't Ghan (2013) show that the change in cloud radiative effect is not a good estimate of ERFaci because it contains pieces of ERFaci and ERFari?

l. 130: It might be worth pointing out that the Heyn et al. (2017) behavior *is* present in the zonal mean distribution (wherever SW ERFaci becomes stronger [weaker], LW ERFaci also becomes stronger [weaker]), but not in the global mean.

Fig. 3: The legend should say what the aggregation is, i.e., are the box and whiskers calculated based on monthly mean grid boxes? Also, in my mind, "susceptibility" implies susceptibility to a measure of aerosol; I would call the LWP, RWP, and SWP changes $\Delta$LWP etc.

l. 190: See my comment about retuning above. For example, in Mülmenstädt et

al. (2020), https://doi.org/10.1126/sciadv.aaz6433, we found that ERFaci is fairly insensitive to the cloud droplet number exponent but very sensitive to the liquid water mixing ratio exponent and the overall normalization in the Khairoutdinov and Kogan (2000) autoconversion scheme. If the retuning strategy for the change in $N_c$ exponent involves changing other parts of the autoconversion, that may result in an overly strong apparent ERFaci change. Of course, which parameters ERFaci is sensitive will vary between models.

l. 210: Is this list complete? E3SM has prognostic snow (Rasch et al., 2019), and I believe GISS Model E3 does too. HadGEM3 may do so as well.

---

## Author Comment (AC1) · 26 Aug 2020

**Response to Reviewer #1 of acp-2020-232**

Dear Reviewer #1,

Thank you very much for taking your time to review our paper. We think that your comments greatly help improve the manuscript. We have revised the manuscript according to your comments as explained below with point-by-point responses to your comments. We hope that the revision is enough to address your comments to make the manuscript now acceptable for publication in *ACP*.

**[RC]**: *Referee comment*
**[AC]**: **Author comment**

**Reviewer #1:**
**General comments:**
**[RC]** *This manuscript demonstrates the impact of falling hydrometeors on the effective radiative forcing of aerosol-cloud interactions in MIROC6 by modifying their microphysics from diagnostic precipitation to prognostic precipitation. The authors find that the large deviation in ERF(aci) in the old scheme is significantly improved as the magnitude of ERF(aci) is reduced. A series of sensitivity tests reveal that prognostic snow has substantial impact on both shortwave and longwave forcing induced by aerosols. This is an excellent modeling work, particularly that the impacts of many physical processes relating to aerosol-cloud interactions are quantified, as illustrated in Figure 4. However, there are some issues need to be clarified before accepted for publication.*

**[AC]** We would like to thank referee #1 for his/her positive comments and suggestions. We agree with the referee's comment that the manuscript needs more discussion about the physical mechanisms behind the snow-induced buffering ACI. We have added a more detailed description of the structural difference between DIAG and PROG, the roles of aerosol scavenging processes, and limitation of the present study, as shown in **[AC1]**, **[AC2]**, **[AC3]**, **[AC4]**, **[AC5]**, and **[AC16]** below.

In the revised manuscript, we have improved the method for estimating ERFaci in MIROC through providing "clean-sky ERFaci" based on Ghan (2013) to preclude contamination by aerosol-radiation interaction in cloudy-sky condition, according to the other referee's comment (Figures R2–R4; **[AC6]**). Furthermore, we have added a new estimate of multi-satellite ERFaci which considers the retrieval error based on Ma et al. (2018), as shown in Figure R3.

The reply and corrections on individual comments are shown below.

**Major concerns:**
**[RC1]** *ERF(aci) is determined not only by changes in optical properties of cloud and precipitating particles, but also by changes in the cloud cover, particularly in GCM. In the text, the authors only show that the simulated cloud cover is "in good agreement" with the observation (although I think underestimation of high clouds is quite severe). However, the aerosol-induced change in cloud cover is not discussed at all.*

**[AC1]** Thank you for your important comment. This study does not discuss the contribution of changes in cloud cover to ERFaci, but it can also contribute to ERFaci although the magnitude might be relatively minor compared to the CLWP adjustment (e.g., Mülmenstädt et al., 2019; Gryspeerdt et al., 2020). The ERFaci correlates the PD minus PI change in CLWP robustly than that in cloud cover (Gettelman, 2015) as shown in Figure R1 because aerosol effects are directly linked to CLWP rather than cloud fraction by modifying the mass conversion rate from cloud water to rainwater that itself relates to the treatment of precipitation (i.e., DIAG vs PROG). However, there might be a feedback from precipitation treatment onto cloud fraction and thus it is nevertheless important in future studies to quantify the impact of precipitation modeling on relative contributions of the CLWP adjustment and cloud fraction adjustment through decomposing the forcing into the two components (Mülmenstädt et al., 2019). We have added brief arguments of this issue as follows (Section 5, Line 239): "This study primarily focused on ERFaci sensitivities to the CLWP adjustment rather than cloud fraction

adjustment, because aerosol effects are directly linked to the CLWP change through the modification of the mass conversion rate from cloud water to rainwater that itself relates to the treatment of precipitation (i.e., DIAG vs PROG). However, it is important in future studies to separating the ACI into the Twomey forcing and rapid adjustments of CLWP and cloud fraction (e.g., Goren and Rosenfeld, 2014; Mülmenstädt et al., 2019) for better understanding how the treatment of precipitation influences micro- and macroscopic cloud properties (Michibata and Suzuki, 2020), which relates to the fundamental inter-model spread in ERFaci (Gryspeerdt et al. 2020; Bellouin et al., 2020).".

We have also modified the description of the modeled cloud cover (Section 2.2) according to the comment.

[Figure]

**Figure R1.** The relation between clean-sky ERFaci and **(left)** change in CLWP and **(right)** change in total cloud cover (TCC) from preindustrial to present-day conditions. Plots in red are tests of different precipitation framework, plots in blue are liquid autoconversion tests, plots in cyan are ice microphysics tests, and plots in green are tests for freezing processes.

**[RC2]** *From this manuscript, it is not clear to me how the aerosol scavenging process works in MIROC6 DIAG or PROG version. I suppose that PROG can simulate aerosol washout more realistic with falling rain and snow, and both ERF(aci) and ERF(ari) could be influenced.*

**[AC2]** According to our parallel analysis that was focused on sensitivities of ERFaci to the scavenging process, both wet-scavenging and coalescence scavenging are found to contribute to decreasing the magnitude of ERFaci. The enhancement of accretion and thus precipitation in the PROG simulation results in a more efficient scavenging process, which weakens the magnitude of the ACI (accretion-driven buffering mechanisms) as detailed in our recent publication (Michibata and Suzuki, 2020). We have added this argument as follows (Line 199): "Although the ERFaci variations with changing liquid and ice microphysical processes do not reach the difference of ERFaci between the DIAG and PROG (i.e., 54%), both wet-scavenging of aerosols and coalescence scavenging of cloud droplets also contribute to the ACI reduction (McCoy et al., 2020) due to the accretion-driven buffering mechanisms (Michibata and Suzuki, 2020), which should explain the remaining part of the ERFaci difference.".

**[RC3]** *This manuscript primarily focuses on the SW, while there is very little discussion about LW. The change in ERF(aci)-LW only attributes to "adjustment induced by snow together with its radiative effect" (Line 139). It is way too brief. The authors should elaborate the physical processes related to the "adjustment".*

**[AC3]** The following discussion has been added (Line 141): "The LW ERFaci is significant over the Indian Ocean and Southeast Asia (not shown), which is also similar to the other model, CAM5-MARC-ARG (Grandey et al., 2018). This is attributable to the increased ice nuclei (IN) due to biomass burning for example, partly supporting the convective invigoration (Rosenfeld et al., 2014) although GCMs do not have enough capability to resolve the convective cloud systems. The increased IN results in a faster glaciation and thus enhances snowfall due to the WBF process (i.e., glaciation

indirect effect). These mixed- and ice-phase microphysical processes are more elaborated in the PROG scheme, and the associated LW change induced by snow is incorporated only in the PROG, which contributes to the higher LW ERFaci across the globe.".

[RC4] *The authors claim that significant reduction in ERF(aci) well corresponds to snow water path (SWP) geographically (Fig. S3). It seems not true for me, particularly over the Southern Ocean. In contrast, the correlation between CLWP (Fig. S4) and ERF(aci) (Fig. 1) looks very high. It may imply that the change in ERF(aci) is primarily determined by CLWP change, and its relation with SWP is not so important.*

[AC4] The ERFaci primarily correlates with the change in CLWP through PI to PD as pointed by the reviewer. Since anthropogenic aerosols are very limited over the remote ocean, particularly over the Southern Ocean, the ERFaci is somewhat noisy and insignificant. What the authors are describing here is that the geographical difference of ERFaci between the DIAG and PROG is similar to the SWP distribution over East Asia, Europe, and North America where anthropogenic pollution dominates. The CLWP should increase with aerosols for both DIAG and PROG due to the cloud lifetime effect, but the significant part of increased cloud water contributes to the source of rain and snow (Fig. 3) when the model prognoses precipitating hydrometeors, and the latter plays a key role of the buffering of ACI (Fig. 4) over these anthropogenic regions.

[RC5] *The sequence of inference in Section 4 is complicated and not straightforward to me. The authors fist state that SWP susceptibility is correlated to LWP susceptibility (which I think a little weak) and propose a hypothesis that riming is important to reduce the optical depth of clouds. However, the sensitivity test clearly reveal the importance of riming (Fig. 4). That is, the sensitivity experiment should be considered as the basis of speculation, not the supporting evidence to the hypothesis.*

[AC5] The hypothesis that the snow-induced buffering of ACI is mainly due to the riming process is one of the interpretations based on sensitivity tests and previous studies (e.g., Lohmann, 2017) as pointed by the reviewer. Since the mixed-phase clouds are influenced by the larger number of microphysical processes than warm-phase clouds, constraining the physical mechanisms of the buffering may be still difficult due to the limitation of isolating feedbacks from multiple microphysical processes. This needs a theoretical approach and idealized process modeling (Glassmeier and Lohmann, 2016; Glassmeier et al., 2019) in addition to a GCM study, which are our important future work beyond the present study. In the revised manuscript, we have added these arguments about the relevant works to solidify the process-level understanding of snow-induced buffering hypothesis.

Section 5 (Line 238): "Furthermore, a theoretical approach (Glassmeier and Lohmann, 2016) and idealized process modeling (Glassmeier et al., 2019) are also required urgently to solidify the process-level understanding of snow-induced buffering hypothesis, which are our important future work beyond the present study."

**Minor points:**

[RC6] *It seems to me that Figure 4 only consider the changes in ERF(aci)-SW because snow radiation is to increase ERF here. If it is the case, the figure caption and the description in the X-axis need to be more clear.*

[AC6] Figure 4 considers the net changes in ERFaci, but the sign of the "snow radiation" in the initial submission was wrong. This should be positive (+12.1%) because the inclusion of snow radiative effect enhances the magnitude of the ACI from -0.71 W m-2 (PROG OFF/SnwRad) to -0.79 W m-2 (PROG ON/SnwRad). The homogeneous freezing was also incorrect in sign, but the other items were all appropriate.

In the revised manuscript, the values of ERFaci are updated by considering the "clean-sky" condition based on Ghan (2013) as suggested by the other reviewer #2. Please be aware that Figures 1, 2, and 4 have been changed in this regard (see Figures R2–R4 below), but the conclusion of this paper does not change.

[Figure]

**Figure R2** (original Figure 1). Geographical distribution of the annual mean clean-sky ERFaci for the **(a)** DIAG and **(b)** PROG precipitation schemes. ERFaci is decomposed into (red) longwave and (blue) shortwave components in the **(c)** zonal mean field for the (dashed) DIAG and (solid) PROG schemes.

[Figure]

**Figure R3** (original Figure 2). ERFaci (Net ERFaci in green; LW ERFaci in red; SW ERFaci in blue) simulated from MIROC6 with different precipitation frameworks. The Net ERFaci values from observation-based studies (Chen et al., 2014; Christensen et al., 2016, 2017; Douglas and L'Ecuyer, 2020) and their probable range (box-whisker) calculated by correcting the effect of retrieval limitations (Michibata and Suzuki, 2020) based on Ma et al. (2018) are also shown. Error bars and plots in MIROC6 represent the minimum/maximum and median of the interannual variability, respectively. Shaded in light-green is the uncertainty range of ERFaci estimated from IPCC AR5 (Boucher et al., 2013). The prognostic rain with diagnostic snow scheme is denoted as "PRDS". The sensitivity experiment without snow radiative effects is denoted as "OFF/SnwRad".

[Figure]

**Figure R4** (original Figure 4). Percentage change of global annual mean clean-sky ERFaci in response to (red) the precipitation treatment, (blue) liquid microphysics, (cyan) ice microphysics, and (green) nucleation of new ice particles due to freezing. Error bars represent the minimum and maximum range for each component considered in this study.

**[RC7]** *Line 87: What is the standard time step in MIROC6 as the sub-time step is 60 s?*
**[AC7]** The standard model timestep is 12 min in MIROC6 used in this study. We have added the information in the revised manuscript (Line 97).

**[RC8]** *Line 95: Does it mean PD and PI simulations use the same SST and sea ice?*
**[AC8]** Yes, PI simulations refer only to the aerosol emissions, and greenhouse gases and SSTs remain at the PD conditions. The differences represent only the aerosol emissions, and is the standard way for diagnosing aerosol ERFs.

**[RC9]** *Line 108: WBF should be spelled out.*
**[AC9]** We have defined the term here with relevant references (Wegener, 1911; Bergeron, 1935; Findeisen, 1938) in the revised manuscript.

**[RC10]** *Line 110: ". . . were returned. . ." Does it mean the simulations without TOA balance are not used in analysis?*
**[AC10]** Yes, all the experiments used in this study are satisfied with the imbalance of TOA radiation within 1.0 W m-2. Model tuning was conducted by modifying scale factor for accretion rate but not autoconversion for warm rain process, because the latter can influence the magnitude of ACI due to the direct relation to droplet number (Michibata and Takemura, 2015; Jing et al., 2019) and thus the precipitation initiation (Mülmenstädt et al., 2020). This is effective for modifying SW radiation, but if needed, cloud ice and snow processes were also tuned for modifying LW radiation by changing scale factors for the fall speed of hydrometeors, which may be uninfluential on ERFaci because they are not involved in the hydrometeor number densities. These tuning strategies have been added in the revised manuscript according to the comments also by the other reviewer.

**[RC11]** *Line 123: This argument regarding ERF(ari) and ERF(aci) is quite indirect. Why not just show the values of these two terms?*

**[AC11]** Since the ARI and ACI are related to each other, it is difficult to separate into ERFari and ERFaci. We have modified the value of ERFaci throughout the manuscript by replacing clean-sky ERFaci as shown in **[AC6]**.

**[RC12]** *Line 126: "i.e., close to zero" Its meaning is ambiguous.*

**[AC12]** This has been removed, thanks.

**[RC13]** *Line 144: "only the latter process. . ." Should be "former"?*

**[AC13]** No, this sentence is correct. The autoconversion process depends on $N_c$ but the accretion process does not.

**[RC14]** *Figure 3: the description in X-axis should be CLWP instead of LWP.*

**[AC14]** We have modified the X-label and caption of Figure 3 as shown below (Figure R5).

[Figure]

**Figure R5** (original Figure 3). Relationship between the change in annual mean CLWP and that in annual mean **(a)** RWP and **(b)** SWP, from the change in aerosols from PI to PD conditions, simulated using the PROG scheme. Box-whisker plots represent the 10th, 25th, 50th (black "+"), 75th, and 90th percentiles of the data within each bin based on the annual mean. Plots in red show the mean. The correlation coefficient (*r*) is given in the figure.

**[RC15]** *Line 174: "water mass suspended in the atmosphere" Does it mean cloud only?*

**[AC15]** Yes, this means cloud water. We have modified this sentence.

**[RC16]** *Line 175: If the total mass is the same, this argument holds. However, in this simulation, the total mass in PROG is larger than DIAG, and then the optical thickness of cloud+precipitation in PROG can be larger than that of cloud only in DIAG.*

**[AC16]** Thank you for the important comment. The total mass of cloud and precipitation hydrometeors is somewhat different between the DIAG and PROG as pointed by the reviewer and our previous study (Michibata et al., 2019). However, this does not primarily determine the ERFaci, and the magnitude of the ERFaci is determined by the changes of cloud and precipitation hydrometeors through PI to PD. If models prognose precipitating hydrometeors as well, the PD-PI change of clouds is shared with precipitating hydrometeors which are less sensitive to SW radiation, which weakens the cooling SW effect significantly. This mechanism can therefore be robust even if the total mass is not the same between the DIAG and PROG.

**References:**

Bergeron, T. (1935). On the physics of cloud and precipitation. Proces verbaux de l'association de meteorologie, intl. union of geod. and geophys., paris (pp. 156–178).

Bellouin, N., Quaas, J., Gryspeerdt, E., Kinne, S., Stier, P.,Watson-Parris, D., et al. (2020). Bounding global aerosol radiative forcing of climate change. Reviews of Geophysics, 58, e2019RG000660. https://doi.org/10.1029/2019RG000660

Douglas, A., and L'Ecuyer, T. (2020). Quantifying cloud adjustments and the radiative forcing due to aerosol-cloud interactions in satellite observations of warm marine clouds. Atmospheric Chemistry and Physics, 20, 6225–6241.

Findeisen, Z. (1938). Kolloid meteorologische vorgange bei neiderschlags-bildung. Meteorologische Zeitschrift, 55, 121–133

Gettelman, A. (2015). Putting the clouds back in aerosol–cloud interactions. Atmospheric Chemistry and Physics, 15, 12,397–12,411. https:// doi.org/10.5194/acp-15-12397-2015

Ghan, S. J. (2013). Technical note: Estimating aerosol effects on cloud radiative forcing. Atmospheric Chemistry and Physics, 13, 9971–9974. doi:10.5194/acp-13-9971-2013

Glassmeier, F., and U. Lohmann (2016), Constraining precipitation susceptibility of warm, ice- and mixed-phase clouds with microphysical equations, Journal of the Atmospheric Science, 73, 5003–5023, doi:10.1175/JAS-D-16-0008.1.

Glassmeier, F., Hoffmann, F., Johnson, J. S., Yamaguchi, T., Carslaw, K. S., and Feingold, G. (2019). An emulator approach to stratocumulus susceptibility, Atmos. Chem. Phys., 19, 10191–10203, https://doi.org/10.5194/acp-19-10191-2019.

Goren, T., and Rosenfeld, D. (2014). Decomposing aerosol cloud radiative effects into cloud cover, liquid water path and Twomey components in marine stratocumulus, Atmospheric Research, 138, 378–393.

Grandey B. S., et al. (2018). Effective radiative forcing in the aerosol–climate model CAM5.3-MARC-ARG, Atmospheric Chemistry and Physics, 18, 15783–15810

Gryspeerdt, E., Mülmenstädt, J., Gettelman, A., Malavelle, F. F., Morrison, H., Neubauer, D., et al. (2020). Surprising similarities in model and observational aerosol radiative forcing estimates. Atmospheric Chemistry and Physics, 20, 613–623. https://doi.org/10.5194/ acp-20-613-2020

Jing, X., Suzuki, K., and Michibata, T. (2019). The key role of warm rain parameterization in determining the aerosol indirect effect in a global climate model. Journal of Climate, 32, 4409–4430.

Mülmenstädt, J., et al. (2020). Reducing the aerosol forcing uncertainty using observational constraints on warm rain processes, Science Advances, 6, eaaz6433.

Mülmenstädt, J., Gryspeerdt, E., Salzmann, M., Ma, P.-L., Dipu, S., and Quaas, J. (2019). Separating radiative forcing by aerosol–cloud interactions and fast cloud adjustments in the ECHAM-HAMMOZ aerosol-climate model using the method of partial radiative perturbations. Atmospheric Chemistry and Physics, 19, 15,415–15,429. https://doi.org/10.5194/acp-19-15415

Ma, P. L., Rasch, P. J., Chepfer, H., Winker, D. M., and Ghan, S. J. (2018). Observational constraint on cloud susceptibility weakened by aerosol retrieval limitations. Nature Communications, 9, 2640. doi:10.1038/s41467-018-05028-4

McCoy, D., Field, P., Gordon, H., Elsaesser, G., and Grosvenor, D. (2020). Untangling causality in midlatitude aerosol-cloud adjustments. Atmospheric Chemistry and Physics, 20, 4085–4103. https://doi.org/10.5194/acp-20-4085-2020

Michibata, T., and Suzuki, K. (2020). Reconciling compensating errors between precipitation constraints and the energy budget in a climate model. Geophysical Research Letters, 47, e2020GL088340. https://doi.org/10.1029/2020GL088340

Michibata, T., and Takemura, T. (2015). Evaluation of autoconversion schemes in a single model framework with satellite observations. Journal of Geophysical Research: Atmospheres, 120, 9570–9590.

Wegener, A., 1911: Thermodynamik der Atmosphäre. J. A. Barth, 331 pp.

Thank you very much again for reviewing our paper.

Sincerely yours,

Takuro Michibata

---

## Author Comment (AC2) · 26 Aug 2020

**Response to Reviewer #2 of acp-2020-232**

Dear Reviewer #2 (Johannes Mülmenstädt),

Thank you very much for taking your time to review our paper. We think that your comments greatly help improve the manuscript. We have revised the manuscript according to your comments as explained below with point-by-point responses to your comments. We hope that the revision is enough to address your comments to make the manuscript now acceptable for publication in *ACP*.

**[RC]**: *Referee comment*
**[AC]**: **Author comment**

**Reviewer #2 (Johannes Mülmenstädt):**

**[RC]** *I have reviewed "Snow-induced buffering in aerosol–cloud interactions" by Takuro Michibata et al. The authors present an interesting set of sensitivity studies that show that prognostic snowfall in their GCM strongly reduces ERFaci compared to diagnostic snowfall, in large part because the longer residence time of snow leads to greater collection of cloud water by snow, which reduces the relative importance of warm phase ACI. Based on my own work, I think this is a plausible mechanism. There are a few potential weak links in the argument, which I will point out below. I don't think those should hold up publication of this potentially very useful result; after all, no paper is ever the final word on any topic. I recommend minor revisions to clarify the points I list below.*

**[AC]** We would like to thank Johannes Mülmenstädt for his carefully reading our manuscript and for giving insightful comments. We have revised the manuscript according to the referee comments.
In the revised manuscript, we have added a new estimate of multi-satellite ERFaci which considers the retrieval error based on Ma et al. (2018), as shown in Figure R3 (see **[AC3]** below). Furthermore, we have improved the method for estimating ERFaci in MIROC through providing "clean-sky ERFaci" based on Ghan (2013) to preclude contamination by aerosol-radiation interaction in cloudy-sky condition (see **[AC4]** for details).
The reply and corrections on individual comments are shown below.

**Major points**

**[RC1]** *l. 110: The tuning strategy needs to be described in more detail. The worry with retuning is that the ERFaci difference may not be due to the change that was intentionally made, but due to the retuning.*

**[AC1]** Thank you for sharing your recent study which documents the link between tuning processes and ERFaci as commented in **[RC7]**. In our study, we primarily tuned the warm rain efficiency by modifying scale factor for accretion rate but not autoconversion because the latter can influence the magnitude of ACI due to the direct relation to droplet number (Michibata and Takemura, 2015; Jing et al., 2019) and thus the precipitation initiation (Mülmenstädt et al., 2020). This is effective for modifying SW radiation, but if needed, cloud ice and snow processes were also tuned for modifying LW radiation by changing scale factors for the fall speed of hydrometeors, which may be uninfluential on ERFaci because they are not involved directly in the hydrometeor number densities. We have added these statements in this paragraph.

**[RC2]** *l. 170 ff.: This is a question rather than a comment. From the conclusion of the paper, I would have expected the relationship between dLWP and dSWP to be the opposite – that when there is more snow, it would lead to more efficient removal of liquid cloud water. Would you mind making these plots separately for supercooled and nonsupercooled water?*

**[AC2]** This is a very important point. Considering the instantaneous response, the more snow should result in more efficient removal of the underlying cloud water due to collection processes, as the reviewer pointed. However, this does not always mean that the relationship between dCLWP and dSWP should be negative because the increased CLWP with aerosols (dCLWP > 0) contributes to additional sources of rain and snow (dRWP > 0; dSWP > 0), which would make the positive relationship in the monthly mean scale. Although the increased response of CLWP with aerosols through PI to PD is basically the same for both DIAG and PROG due to the cloud lifetime effect, the precipitation-driven collection of cloud droplets in PROG buffers the dCLWP (not mean negative) and thus the magnitude of ACI. The significant buffering of ACI can be seen when the model prognoses snow (Fig. 4) particularly over anthropogenic regions where snow is abundant, which does not conflict the positive relationship between dCLWP and dSWP.

To understand how the dCLWP-dSWP relation depends on the cloud regime, we further looked at the plot for supercooled and nonsupercooled regimes separately (Figure R6). The relationship between dCLWP and dSWP is more robust in the supercooled regime than in the nonsupercooled regime, implying that the mixed-phase cloud microphysics is a key driver to the snow-induced ACI buffering. Although we did not obtain a negative correlation between dCLWP and dSWP from this analysis, a theoretical approach and idealized process modeling should be required for future study to solidify the process-level understanding of snow-induced buffering hypothesis.

We have added the following sentence in the revised manuscript (Line 238): "Furthermore, a theoretical approach (Glassmeier and Lohmann, 2016) and idealized process modeling (Glassmeier et al., 2019) are also required urgently to solidify the process-level understanding of snow-induced buffering hypothesis, which are our important future work beyond the present study."

[Figure]

**Figure R6**. Relationship between the change in annual mean CLWP (nonsupercooled in blue and supercooled in cyan) and that in annual mean SWP, from the change in aerosols from PI to PD conditions, simulated using the PROG scheme. The correlation coefficients (*r*) are given in the figure.

**Minor points**

**[RC3]** *l. 39 ff.: I don't think this argument is logically consistent. First, the authors say ERFaci in GCMs is "too negative" compared to satellite studies (I would prefer "more negative", since satellite studies have their own problems). But then they cite a (problematic) satellite study with a very negative SW ERFaci to argue that the problem is with the models' LW ERFaci. The rest of the paragraph is fine, but I would suggest removing the first two sentences.*

**[AC3]** Yes, satellite retrievals also include uncertainties which may underestimate ERFaci because satellites undersample the weak-aerosol regime, where the cloud sensitivity to aerosol is largest (Ma et al., 2018). In the revised manuscript, we have added multi-satellite ERFaci which considers the retrieval errors based on Ma et al. (2018) as shown in Michibata and Suzuki (2020), but the DIAG scheme still shows more negative ERFaci (Figure R3). This implies that there might be unknown compensating aerosol warming effects that are missing in current GCMs, possibly through mixed-phase clouds (Lohmann and Hoose, 2009).

We therefore have remained the sentences but modified slightly as follows (Lines 39-42): "As a consequence of the challenges described above, GCMs tend to show more negative ERFaci than that inferred from satellite retrievals (Quaas et al., 2009; Chen et al., 2014) even though retrieval errors (Ma et al., 2018) are considered (Michibata and Suzuki, 2020). This suggests that current GCMs may be missing a compensating warming effect caused by aerosols.".

**[RC4]** *l. 111: Didn't Ghan (2013) show that the change in cloud radiative effect is not a good estimate of ERFaci because it contains pieces of ERFaci and ERFari?*
**[AC4]** In the revised manuscript, we have improved the method for estimating ERFaci in MIROC through providing "clean-sky ERFaci" based on Ghan (2013) to preclude contamination by aerosol-radiation interaction in cloudy-sky condition. This revision changes Figures 1, 2, and 4. We have also added a new estimate of multi-satellite ERFaci which considers the retrieval error based on Ma et al. (2018), as presented in Michibata and Suzuki (2020) (please see also **[AC3]** above).

**[RC5]** *l. 130: It might be worth pointing out that the Heyn et al. (2017) behavior is present in the zonal mean distribution (wherever SW ERFaci becomes stronger [weaker], LW ERFaci also becomes stronger [weaker]), but not in the global mean.*
**[AC5]** The following sentence has been added in this paragraph (Line 128): "The zonal distribution shows that stronger (weaker) LW ERFaci accompanies stronger (weaker) SW ERFaci, which is in line with Heyn et al. (2017).".

**[RC6]** *Fig. 3: The legend should say what the aggregation is, i.e., are the box and whiskers calculated based on monthly mean grid boxes? Also, in my mind, "susceptibility" implies susceptibility to a measure of aerosol; I would call the LWP, RWP, and SWP changes dLWP etc.*
**[AC6]** The following explanation has been added in the caption: "Box-whisker plots represent the 10th, 25th, 50th (black "+"), 75th, and 90th percentiles of the data within each bin based on the annual mean.". The figure legend uses dCLWP, dRWP, and dSWP, instead of "susceptibility".

**[RC7]** *l. 190: See my comment about retuning above. For example, in Mülmenstädt et. al. (2020), https://doi.org/10.1126/sciadv.aaz6433, we found that ERFaci is fairly insensitive to the cloud droplet number exponent but very sensitive to the liquid water mixing ratio exponent and the overall normalization in the Khairoutdinov and Kogan (2000) autoconversion scheme. If the retuning strategy for the change in Nc exponent involves changing other parts of the autoconversion, that may result in an overly strong apparent ERFaci change. Of course, which parameters ERFaci is sensitive will vary between models.*
**[AC7]** Thank you for your very important comment. We have added a more detailed description of the tuning (Section 2.2) as answered in **[AC1]**. We have also modified the sentence citing the suggested work.

**[RC8]** *l. 210: Is this list complete? E3SM has prognostic snow (Rasch et al., 2019), and I believe GISS Model E3 does too. HadGEM3 may do so as well.*
**[AC8]** Thank you for the information. The authors have contacted several modeling centers to know the latest model spec on the treatment of precipitation. Some replies suggested that now more GCMs include two-moment prognostic precipitation with snow radiative effect (e.g., E3SM, GISS-Model E3). In the revised paper (Lines 210 and 230), this sentence has been slightly changed citing a relevant paper (Li et al., 2020) which overviews the latest model status in CMIP6.

**References:**

Chen, Y.-C., Christensen, M. W., Stephens, G. L., and Seinfeld, J. H. (2014). Satellite-based estimate of global aerosol-cloud radiative forcing by marine warm clouds. Nature Geoscience, 7, 643–646.

Ghan, S. J. (2013). Technical note: Estimating aerosol effects on cloud radiative forcing. Atmospheric Chemistry and Physics, 13, 9971–9974. doi:10.5194/acp-13-9971-2013

Glassmeier, F., and U. Lohmann (2016), Constraining precipitation susceptibility of warm, ice- and mixed-phase clouds with microphysical equations, Journal of the Atmospheric Science, 73, 5003–5023, doi:10.1175/JAS-D-16-0008.1.

Glassmeier, F., Hoffmann, F., Johnson, J. S., Yamaguchi, T., Carslaw, K. S., and Feingold, G. (2019). An emulator approach to stratocumulus susceptibility, Atmos. Chem. Phys., 19, 10191–10203, https://doi.org/10.5194/acp-19-10191-2019.

Heyn I, Block K, Mulmenstadt J, Gryspeerdt E, Kuhne P, Salzmann M, Quaas J. Is the IPCC AR5 estimate of the aerosol effective radiative forcing too weak? (2017). Geophys Res Lett.

Jing, X., Suzuki, K., and Michibata, T. (2019). The key role of warm rain parameterization in determining the aerosol indirect effect in a global climate model. Journal of Climate, 32, 4409–4430.

Li, J. L. Frank et al. (2020). An Overview of CMIP5 and CMIP6 Simulated Cloud Ice, Radiation Fields, Surface Wind Stress, Sea Surface Temperatures, and Precipitation Over Tropical and Subtropical Oceans. Journal of Geophysical Research, 125, e2020JD032848

Lohmann, U. and Hoose, C. (2009). Sensitivity studies of different aerosol indirect effects in mixed-phase clouds, Atmospheric Chemistry and Physics, 9, 8917–8934.

Mülmenstädt, J., et al. (2020). Reducing the aerosol forcing uncertainty using observational constraints on warm rain processes, Science Advances, 6, eaaz6433.

Ma, P. L., Rasch, P. J., Chepfer, H., Winker, D. M., and Ghan, S. J. (2018). Observational constraint on cloud susceptibility weakened by aerosol retrieval limitations. Nature Communications, 9, 2640. doi:10.1038/s41467-018-05028-4

Michibata, T., and Suzuki, K. (2020). Reconciling compensating errors between precipitation constraints and the energy budget in a climate model. Geophysical Research Letters, 47, e2020GL088340. https://doi.org/10.1029/2020GL088340

Quaas, J., Ming, Y., Menon, S., Takemura, T., Wang, M., Penner, J. E., et al. (2009). Aerosol indirect effects general circulation model intercomparison and evaluation with satellite data. Atmospheric Chemistry and Physics, 9, 8697–8717. https://doi.org/10.5194/ acp-9-8697-2009

Rasch, P. J., Xie, S., Ma, P.-L., Lin, W., Wang, H., Tang, Q., et al. (2019). An overview of the atmospheric component of the Energy Exascale Earth System Model. Journal of Advances in Modeling Earth Systems, 11(8), 2377–2411. https://doi.org/10.1029/2019MS001629

Thank you very much again for reviewing our paper.

Sincerely yours,

Takuro Michibata